# Did the Addition of Olive Cakes Obtained by Different Methods of Oil Extraction in the Finishing Diet of Bísaro Pigs Affect the Volatile Compounds and Sensory Characteristics of Dry-Cured Loin and “Cachaço”?

**DOI:** 10.3390/foods12132499

**Published:** 2023-06-27

**Authors:** Ana Leite, Lia Vasconcelos, Iasmin Ferreira, Rubén Domínguez, Mirian Pateiro, Sandra Rodrigues, Etelvina Pereira, Paulo C. B. Campagnol, José Angel Pérez-Alvarez, José M. Lorenzo, Alfredo Teixeira

**Affiliations:** 1Centro de Investigação de Montanha (CIMO), Instituto Politécnico de Bragança, Campus de Santa Apolónia, 5300-253 Bragança, Portugal; anaisabel.leite@ipb.pt (A.L.); lia.vasconcelos@ipb.pt (L.V.); iasmin@ipb.pt (I.F.); srodrigues@ipb.pt (S.R.); etelvina@ipb.pt (E.P.); 2Laboratório para a Sustentabilidade e Tecnologia em Regiões de Montanha, Instituto Politécnico de Bragança, Campus de Santa Apolónia, 5300-253 Bragança, Portugal; 3Área de Tecnoloxía dos Alimentos, Facultade de Ciencias de Ourense, Universidade de Vigo, 32004 Ourense, Spain; jmlorenzo@ceteca.net; 4Centro Tecnológico de la Carne de Galicia, Avd. Galicia N° 4, Parque Tecnológico de Galicia, San Cibrao das Viñas, 32900 Ourense, Spain; rubendominguez@ceteca.net (R.D.); mirianpateiro@ceteca.net (M.P.); 5Departamento de Tecnologia e Ciência de Alimentos, Universidade Federal de Santa Maria, Santa Maria 97105-900, Brazil; paulocampagnol@gmail.com; 6Departamento de Tecnologia Agro-Alimentar, Escuela Superior Politécnica de Orihuela, Universidad Miguel Hernández, 03312 Alicante, Spain; ja.perez@umh.es

**Keywords:** Bísaro breed, dry-cured products, olive cake, volatile compounds, sensory

## Abstract

This study was conducted to determine the effects of different types of olive cake in the basal diet of Bísaro pigs on the volatile compounds and sensory characteristics of dry-cured loin and “cachaço”. A total of 40 Bísaro breed animals were allocated to four treatments, along with a control group (T1—control, T2—crude olive cake, T3—centrifugation two phases, T4—exhausted, and T5—exhausted with 1% of olive). Various extraction methods (centrifugation, pressing, and exhaustion) were employed for the olive cake used. Furthermore, the extracted olive cake was supplemented with 1% olive oil. Eighty compounds were identified and grouped into eight chemical classes: hydrocarbons, aldehydes, esters, alcohols, ketones, acids, furans, and other compounds. Aldehydes and alcohols were the major groups of compounds, representing 57.06–66.07% and 68.67–75.61% for the loin and “cachaço”, respectively. There were no significant differences between treatments for any of the volatile compounds identified. The major aldehydes were hexanal, heptanal, pentanal, and propanal. These compounds were significantly higher (*p* < 0.001) in “cachaço”. This significant difference between the two types of dry-cured products was directly related to the amount of total fat content. The major alcohols were 2.3-butanediol, 1-octen-3-ol, 1-butanol, 3-methyl, 1-hexanol, benzyl-alcohol, and glycidol. Except for compounds 2,3-butanediol and benzyl-alcohol, the majority in this group were significantly different in terms of the type of dry-cured product. As for the sensory evaluation, for both dry-cured products, the trained tasters did not detect significant differences between the different treatments. The results showed that the olive cake obtained by different methods of oil extraction did not negatively affect the sensory and volatile components of the processed meat products; thus, they maintained their appeal to the consumer.

## 1. Introduction

The Bísaro pig is a native breed from the *Trás-os-Montes* region (Portugal). Despite being one of the most emblematic native Portuguese breeds, it is rare in Portugal. The decline of the breed, which has nearly led to its extinction, can be attributed to various factors, such as shifts in modern society’s dietary preferences towards lean meats and the phenomenon of rural exodus, among others. In the last few decades, we have observed a great valorization of extensive agriculture associated with the production of high-quality products. This paradigm shift has allowed the Bísaro pig breed to assume decisive importance in the rural development of Trás-os-Montes, contributing to maintaining the typical products of the region as well as traditional production systems.

Dry-cured loin, crafted from a premium section of the pork carcass, stands as one of the most popularly consumed items globally. On the other hand, dry-cured “cachaço”, derived from the front portion of the *Longissimus thoracis et lumborum* muscle, remains relatively less renowned on an international. Compared to other pig breeds such as the Iberian breed, processed products from the Bísaro breed do not have the same commercial value or worldwide recognition. The quality of the dry-cured products is attributed to the inherent qualities of the raw materials, including feeding characteristics, the age of the animals, and the breed of the pigs [1].

When considering the utilization of olive byproducts in animal diets, several aspects are examined, including their composition, digestion, degradation, and influence on animal performance and product quality, with a particular emphasis on their fatty acid profile. Studies using olive cake in ruminant diets have promoted satisfactory responses in terms of providing cheap energy and fiber [2]. Depending on the fat content of the olive cake, this product can be used to improve the quality of the fat in the meat of the animals. In the case of Bísaro pigs, the use of olive cake has little influence on the quality of the carcass and meat [3,4]. According to Dominguez et al. [5], the volatile composition of a dry-cured product results from a set of reactions that are crucial in determining their sensory attributes. These reactions consist of the oxidative decomposition of lipids and the degradation reactions of amino acids. It is expected that trends observed in fatty acid profiles in studies already conducted [3,4] will be observed in volatile compound profiles and that even more significant results will be observed. The different compounds derived from the curing process and the addition of ingredients contribute to the formation of a peculiar aroma, characteristic of these products [5]. Paprika and garlic are two spices with very important functions in terms of the taste of dry-cured products due to volatile compounds (mainly terpenes and sulfide compounds); furthermore, they are important antioxidant agents [6]. The curing process generates a set of chemical changes that lead to the development of volatile compounds, mainly esters and furan compounds [7]. These chemical and biochemical changes lead to changes in the taste as well as in the odor of the final product, which influences the flavor and consumers’ acceptability.

Dry-cured loin is a highly appreciated product worldwide. There are multiple studies on its physicochemical and sensory characteristics with respect to Ibérico products [1,6]. These studies allow us to know the Iberian breed with regard to the quality of its carcass, meat, and processed products. The same should happen with other breeds as important as the Ibérico breed, such as the Bísaro breed. Nevertheless, the Bísaro breed is not as well known worldwide as some other breeds. Therefore, the present study addresses the research gap about this breed and its processed products.

On the other hand, as it is known, there are large amounts of waste that are generated during olive processing [8,9]. Since one of the major environmental objectives is the achievement of a circular economy, these residues could become part of the diet of some animals. Studies have been carried out with the inclusion of chestnut in the finishing diets of animals, which altered the content of volatile compounds in the final product (dry-cured ham) [10]. However, studies carried out on lamb meat with the introduction of olive cake and linseed did not produce appreciable changes in the volatile compound of the meat [11].

Thus, the aim of this study was to evaluate the effect of the inclusion of olive cakes obtained by different methods of oil extraction in finishing diets on the volatile compounds of dry-cured loin and “cachaço” from Bísaro pigs.

## 2. Materials and Methods

### 2.1. Animal Management and Diets

To carry out this study, animals of the Bísaro breed (*Sus scrofa*) with a live weight of 100 kg and an average age of 12 months were used. These animals were randomly divided into 5 groups for a total of 40 animals. The groups included a control group (basal diet) and four groups with different olive cakes (in combination with the basal diet): T1—Basal diet and commercial feed; T2—Basal diet + 10% crude olive cake; T3—Basal diet + 10% olive cake in two phases; T4—Basal diet + 10% exhausted olive cake; T5—Basal diet + 10% exhausted olive cake + 1% olive oil). Several types of extraction were used for the olive cake introduced to the diet of Bísaro pigs (centrifuged (T3), pressed (T2), and exhausted (T4)), and the exhausted olive cake was also used with the addition of 1% olive oil (T5). Since exhausted olive cake is more limited in its fat composition, it was necessary to apply 1% olive oil, thus creating yet another treatment (T5). This choice was made because the exhausted cake is easier to transport, has a greater capacity in terms of shelf life, and is available for a longer time. The animals came from a farm (Bísaro Salsicharia Tradicional^®^, Gimonde, Portugal) and were raised in an extensive production system. In total, 40 animals of the Bísaro breed were used. This feed was applied to all groups simultaneously and under the same conditions (feed level was “ad libitum” with an average consumption of 3 kg per day). The base and commercial diets applied to the pigs’ concerns rations were those typically applied to native pigs. In addition to the feed, the animals in this extensive regime had access to horticultural products of the region such as sweet potatoes, radish, turnips, cabbage, and other seasonal fruits. The fatty acid profiles of the diets used were composed at the Meat Technology Center of Galicia, Ourense, Spain. From the analysis of the fatty acid profiles of the different diets, there were no statistically significant differences in the MUFA, PUFA, and SFA contents of the diets incorporating olive cakes in relation to the control diet (T1). However, the T4 group incorporating exhausted olive cake showed fatty acid profiles most similar to those of T1. For SFA, the values ranged from 20.03 to 21.54% for all treatments. As for MUFA, they varied between 28.0 and 42.36%. As for PUFA, the values ranged between 37.61 and 50.46%. C16:0, C18:1n-9, and C18:2n-6 were the major fatty acids of SFA, MUFA, and PUFA, respectively, in the different treatments used for the diets of the Bísaro pigs. The chemical compositions and fatty acid profiles of the meat in Bísaro breed animals with olive cake included in their diets have been analyzed and published previously (see Table 1) [3].

Different types of olive oil cake from different industries that receive olives from all over the northeast of Portugal were used. The experimental trial was conducted in the finishing phase (the last 90 days before slaughtering the animals) by the “Bisolive project” partners (the University of Tras-os-Montes and Alto Douro, Vila Real, Portugal, UTAD and the extensive Bísara breed pig farm). The Bísaro pigs were slaughtered at the Municipal Slaughterhouse of Bragança as described by Álvarez-Rodríguez and Teixeira [12] and in accordance with applicable legislation [13]. In Table 1, we provide the chemical compositions of the diets, as well as the fatty acid profiles associated with the various treatments.

It should be noted that this article is a continuation of other previously published work. The influence of olive pomace on carcass conformation and the chemical composition of meat can be assessed in more detail in another publication [3].

### 2.2. Dry-Cured Bísaro Loin and Cachaço Manufacture

Forty loins and forty “cachaços” from the forty slaughtered animals were used. The curing process was carried out by the company Bísaro, Salsicharia Tradicional, Lda. The dry-cured loin was made from one of the most valuable joints of pork carcass, the lumbar and thoracic parts of the *Longissimus thoracis et lumborum* (LTL) muscle. The dry-cured “cachaço” was made from the proximal part of the *Longissimus thoracis et lumborum* (LTL) muscle from the column’s cervical part that is noticeable beneath the scapula until the fifth thoracic vertebra. The production process for the loin is the same as that for the “cachaço”, only the size and location of the piece varies (vertebral from the end of the joint of the “cachaço” to the last lumbar vertebra). To create this type of product, a series of steps were applied that led to the formation of the cured product. The following ingredients were added: 1.5% of salt, 0.5% of paprika, 0.5% of garlic, and 0.1% of oregano. The total curing time for both products was approximately 60 days.

After extracting the muscles from the animal carcasses, the pieces (loin and “cachaço”) were refrigerated in a chamber between 2 and 5 °C. The next step was the salting and seasoning phase. Before starting this rotating process (for 30 min), all the ingredients were added. When the mixing was complete, the pieces were placed in a refrigeration chamber between 2 and 4 °C for 4 days. The next phase was stuffing into collagen casings. The last phase was the drying and curing phase, in which the most important biochemical changes take place. The temperature and relative humidity changed as the curing time progressed, increasing the temperature and decreasing the relative humidity. In the first 15 days, the cuts were exposed to temperatures between 4 and 8 °C with a relative humidity between 80 and 90%. After this period, the product was exposed to temperatures between 8 and 12 °C with relative humidity between 70 and 80% for another 15 days. Finally, for the last 20 days, the product was exposed to temperatures between 12 and 18 °C with a relative humidity between 60 and 70%. The curing method applied was conducted according to Leite et al. [4].

The influence of olive cake on the physicochemical compositions and fatty acid profiles of dry-cured products (loin and cachaço) can be consulted in detail in a previous article [13].

### 2.3. Volatile Compounds

The volatile compounds of 1 g of the samples were analyzed using the SPME-gas chromatography–mass spectrometry technique (Agilent Technologies, Santa Clara, CA, USA), following the procedure described by Dominguez et al. [5]. The volatile results were expressed as area units per gram of sample (AU × 10^5^/g of the sample).

### 2.4. Sensorial Analysis

The dry-cured loin and “cachaço” samples were evaluated by a trained taste panel. The training was conducted according to the Portuguese Standard [14] in the Sensory Analysis Laboratory at the Polytechnic Institute of Bragança. This panel (made up of nine tasters) was created after recruitment, selection, and training phases. The conditions of the test room where the evaluation took place followed standard guidelines [15]. The light in the room and each booth was white to facilitate evaluation. The panelists were given water and unsalted toasts to cleanse the palate and remove residual flavors at the beginning of the session and between samples. Samples of dry-cured loin and “cachaço” were divided into 1.5 mm thick slices, placed on a plate at room temperature, and evaluated. For both loin and cachaço, two samples of each treatment were evaluated in each session, with 3 sessions per product. Each treatment was evaluated 6 times by each taster. Samples were evaluated for quantitative attributes related to appearance, odor, texture, taste, and flavor. A 9-point scale, in which 1 represented the minimum (very weak intensity) and 9 represented the maximum (very strong intensity), was used for the quantitative attributes, considering a quantitative descriptive analysis.

### 2.5. Statistical Analysis

Data were tested for normal distribution and variance homogeneity by the Shapiro–Wilk test. Then, the effect of product, treatment, and the interaction between product and treatment (data not shown for the latter once all interactions were not significant) on volatile compounds contents was examined using analysis of variance (ANOVA) with the general linear model (GLM) procedure, where these parameters were set as dependent variables and product and treatments were set as fixed effects. The results were given in terms of mean values and standard error of the mean (SEM). When a significant effect (*p* < 0.05) was detected, means were compared using Student’s *t*-test.

Simple means were used to develop a sensory profile for the dry-cured loin and “cachaço”. A non-parametric analysis of variance was performed for the sensory data and treatments were compared by Friedman’s test using SPSS software 22.0 program (IBM SPSS Statistics). Using the XLStat program (Addinsoft, New York, NY, USA), a generalized procrustean analysis (GPA) was used to minimize the differences between assessors, identify agreements between them, and summarize the sets of 3-dimensional data.

## 3. Results and Discussion

### 3.1. Volatile Compounds of Dry-Cured Loin and “Cachaço”

The distinct locations within the same muscle utilized in the production of these two varieties of dry-cured products exerted a significant influence on a considerable portion of the detected volatile organic compounds (VOCs). These compounds were classified into eight chemical classes: hydrocarbons (21), aldehydes (18), esters (13), alcohols (12), ketones (12), acids (4), furans (4), and other compounds (4). According to several authors [16,17], the composition of volatile compounds is attributed to the smoking process, added seasonings, and interactions involving lipids, proteins, carbohydrates catalyzed by microbial enzymes, and oxidative processes.

Aldehydes were the main volatiles found in all treatments (Table 2), representing 34.84–46.45% and 50.77–55.68% of the total VOC in dry-cured loin and “cachaço”, respectively. The aldehydes could be divided into two groups: linear aldehydes and branched aldehydes. The linear aldehydes were derived mainly from lipid oxidation and the branched aldehydes were related to amino acid degradation and proteolysis [3,18,19]. The major aldehydes were hexanal, heptanal, pentanal, and propanal. These compounds were significantly higher (*p* < 0.001) in the dry-cured “cachaço”. This significant difference between the two types of dry-cured products was directly related to the amount of total fat content. According to Leite et al. [4], dry-cured “cachaço” has a considerably higher amount (~45%) of total fat than dry-cured loin (~20%). Linear aldehydes are typical products of lipid oxidation and are responsible for the fat odor. The main linear aldehyde was hexanal, as observed by other authors in other products: polish dry-cured loin [20], dry-fermented deer sausage [21], dry-cured traditional Istrian ham [22], cecina [23], and fermented sausages [24]. Hexanal and pentanal were derived from the oxidation of unsaturated fatty acids, namely through the lipoxygenase pathway (LOX), from linoleic, linolenic, and arachidonic fatty acids, while heptanal was derived from oleic acid [18,20,22,25]. As can be observed in Table 2, hexanal and pentanal contents were not influenced by the treatments used in the animal feed. This fact could be related, as was reported in a previous study [4], to the inclusion of olive cakes, which did not produce changes in the fatty acid profiles of these products; thus, the oxidative degradations could be similar in both. According to Domínguez et al. [5] the hexanals present a rancid aroma at high amounts, while at low amounts, it gives a pleasant grassy aroma. Górska et al. [20] described the hexanal aroma as green, grassy, fatty, rancid strong, unpleasant hot, and nauseating. With this in mind, it is expected that low-medium levels of this aldehyde produce a desirable flavor in the final product, which is related to the “curing” process and appreciated by the consumer.

In addition to these compounds, other important aldehydes (branched and aromatic) were also found in dry-cured loin and “cachaço” of Bísaro pork, namely, 3-methyl-butanal, benzaldehyde, and benzeneacetaldehyde. These compounds were also found in dry-fermented deer sausage [21], dry-cured meat [5], fermented sausages [26], and cecina [23]. The origin of 3-methyl-butanal is the deamination–decarboxylation of the amino acid leucine, whereas benzaldehyde and benzeneacetaldehyde are derived from the Streaker degradation of some amino acids such as leucine or phenylalanine [5,6,19]. In this sense, aromatic aldehydes (benzaldehyde and benzeneacetaldehyde) possess floral, bitter almond notes and rancid and pungent aromas, while 3-methyl-butanal is an important compound for dry-cured products since it has a typical “ripened flavor” [5].

Alcohols were the second major group in the volatile profile (Table 2) and represented between 19.06 and 19.62% and 17.90 and 19.93% of the total VOC in dry-cured loin and “cachaço”, respectively. The major alcohols were 2,3-butanediol, 1-octen-3-ol, 1-butanol, 3-methyl, 1-hexanol, benzyl-alcohol, and glycidol. These compounds were also observed in fermented sausages [5,21,26]. With the exception of compounds 2,3-butanediol and benzyl-alcohol, the majority in this group were significantly different in terms of the type of dry-cured product. As for the different treatments, and in line with what happened with the aldehydes group, there were no significant differences in the major compounds. Of the major compounds in this group of volatiles, 1-octen-3-ol stood out in the dry-cured “cachaço”, while 1-butanol, 3-methyl, 1-hexanol, and glycidol were found in greater proportions in the dry-cured loin. According to other authors [5,19] 1-octen-3-ol, derived from the oxidation process of linoleic acid, is described as a very important compound for contributing a dry-cured aroma to products. According to Leite et al. [4], dry-cured “cachaço” contained a significantly higher linoleic acid content than that observed in dry-cured loin, which may explain the presence of this compound in greater proportions in the dry-cured “cachaço”. The major content of 1-butanol, 3-methyl may have been due to the activity of the microorganisms present in the dry-cured loin. According to Muriel et al. [27], microorganisms can act on butanal, 3-methyl, formed by the degradation of amino acids during proteolysis to produce 1-butanol, 3-methyl.

Four compounds were isolated in the group of acids (Table 2): butanoic acid, 3-methyl; butanoic acid; hexanoic acid; and acetic acid. The acids represented 0.87–1.97% and 3.45–5.80% of the total VOC in dry-cured loin and “cachaço”, respectively. The butanoic, hexanoic, and acetic acid compounds were significantly different (*p* < 0.001) with regard to the type of product. Again, the treatment had no influence on any of the products studied. The highest amounts of butanoic and hexanoic acid were observed in the dry-cured “cachaço”. These compounds have also been found in dry-fermented deer sausages [21], dry-cured loin and shoulders [5], and cecina [5,23]. Butanoic acid, 3-methyl was also found in dry-cured ham [8] and loin [20]. According to some authors [20], butanoic acid, 3-methyl can be generated from leucine by the functions of some Staphylococcus. The aroma of this compound has been described as cheese, feet, fatty, and rancid, and may contribute to the lower overall quality of products [20]. Acetic acid gives notes of ripeness [21]. The most probable origin of hexanoic acid is the carbohydrate fermentation induced by microorganisms such as lactic bacteria and staphylococci [28,29].

For these three groups of volatile compounds, no significant differences (*p* > 0.05) were detected between the treatments. In previous studies [4], the fatty acid fractions obtained for these products were not affected by the introduction of olive cake into animal feed. Considering that some of the mechanisms for the formation of volatile compounds, namely through the lipoxygenase pathway, are conducted at the level of fatty acids, we can conclude that it was possible that there were no differences at the level of the different treatments and their interaction with the type of product.

Hydrocarbons were the third largest compounds found in these types of dry-cured products and represented 13.68–20.37% and 6.78–9.34% of the total VOC in dry-cured loin and “cachaço”, respectively (Table 2). The major hydrocarbons were octane, 2,2,4,4-tetramethyloctane, and heptane. These compounds were significantly higher in dry-cured loin, with no influence of the treatments applied to the animals’ feed. The heptane and octane compounds (linear alkanes) have also been described by other authors in relation to Iberian ham and dry-cured loin [8,16,25,30]. 2,2,4,4-tetramethyloctane (branched alkanes) has also been found in dry-cured products such as ham, salchichón, shoulder, and cecina [5]. The appearance of these branched alkanes, especially methyl hydrocarbons, is related to the activity of molds, which synthesize these compounds as a product of the secondary degradation of triglycerides [31,32].

As for the ketones (Table 2), there were no significant differences between the two types of products and no significant differences between the treatments (*p* > 0.05). The ketones represented 6.79–11.60% and 5.92–8.93% of the total VOC in dry-cured loin and “cachaço”, respectively. The most abundant ketone was acetoin, followed by 2-heptanone. Acetoin is the major ketone in dry-cured products such as loin, salchichón, shoulder, and chorizo [5,21]. As with the dry-cured loin and dry-cured “cachaço” of this study, dry-cured shoulder [5] also demonstrated similar values for the compound 2-heptanone. The origin of ketones can be diverse [5], but according to Pérez-Santaescolástica et al. [32], acetoin is formed through Maillard reactions. According to Sidira et al. [33], acetoin has a buttery, sweet odor, with a very low odor threshold, and contributes to the typical flavor of dry-cured meat products. Regarding 2-heptanone, according to García-González et al. [34], this compound contributes to spicy, blue cheese, and acorn sensory notes, and the release of 2-ketones is related to the oxidative processes of lipids.

Regarding esters (Table 2), butanoic acid, ethyl ester was the most abundant compound in the two types of products. The butanoic acid, 3 methyl, ethyl ester was also found in higher amounts in the dry-cured “cachaço” compared to the dry-cured loin of Bísaro pork. The esters represented 7.93–10.81% and 3.40–5.68% of the total VOC in dry-cured loin and “cachaço”, respectively. For the most abundant compound of this group (butanoic acid, ethyl ester), there were no significant differences between products or treatments. Regarding the butanoic acid, 3 methyl, ethyl ester, this compound showed significant differences between both products, being higher in the dry-cured “cachaço” than in the dry-cured loin. Petričević et al. [35] reported that the main origin of esters in meat products is the esterification of carboxylic acids and alcohols. On the other hand, Marco et al. [36] and Akköse et al. [37] suggested that low molecular weight esters can also be a product of carbohydrate metabolism. In this regard, the action of some microorganisms can promote the enzymatic esterification of fatty acids and alcohols due to their high esterase activity [5].

Four compounds were also found in the furan group (Table 2) (furan, 2-ethyl, 2-n-butyl furan, furan, 2-pentyl, and furan, 2-propyl). The furans represented 1.10–1.72% and 1.53–2.05% of the total VOC in dry-cured loin and “cachaço”, respectively. With the exception of furan, 2-pentyl, the furans were significantly different (*p* < 0.001) with regard to the type of product. Again, the treatment had no influence on any of the products studied. In other studies [5], no furans were found in dry-cured loin. In contrast, the same furans were found in other types of products such as cecina [5] and the dry-cured ham of Celta pigs [8]. Furan, 2-pentyl was also found in dry-fermented deer sausages [21] and the dry-cured loin of Iberian pigs [27]. Furans are described as compounds generated during heating [8]; however, they have already been found in other types of dry-cured products such as ham and loin. According to Ruiz et al. [38], Akköse et al. [37], and Lorenzo et al. [8], this VOC is a compound derived from linolenic and other n-6 fatty acid oxidation reactions. Due to their low odor threshold values, furans play an important role in all meat products. Furan, 2-pentyl, and furan 2-ethyl provide a pleasant aroma [1]

Four compounds were detected in the dry-cured loin and “cachaço” that were not included in the previous groups (Table 2). The other compounds represented 0.94–2.59% and 0.75–1.29% of the dry-cured loin and “cachaço”, respectively. The compounds found were sulfide, allyl methyl, 1H-pryrrole, 3-methyl, 1,3-Benzenediol, monobenzoate, pyrazine, and 2,6-dimethyl. The compound sulfide, allyl methyl was the most expressive in this group, with no significant differences between the two types of products nor between the types of treatments. This compound has been observed in samples of low-fat fermented sausages [26] and chorizo [39]. According to other authors [39], the detection of compounds with sulfur and allyl could be related to the use of garlic as an ingredient. Therefore, they are organosulfur compounds derived from garlic. These compounds are components of the aroma of onion, garlic, and other Allium species [5].

For the groups of volatile compounds mentioned in Table 2, it was also found that there were no significant differences (*p* > 0.05) between the various types of treatments. The introduction of olive cake into the diet of the animals had no impact on the volatile compounds of the products studied. The fact that olive cake from various industries in the region was used may have approximated their composition in terms of volatile compounds, which was also reflected in the final product. Another possible reason why no significant differences were detected between the treatments may be due to the time period for which the diet was applied, proving insufficient for the formation of volatile compounds in the dry-cured products.

### 3.2. Sensory Characteristics

In Figure 1a and Figure 2a, we can observe the average scores obtained in the sensory analyses of the dry-cured loin and “cachaço”. For these products, the panel tasters evaluated 16 quantitative sensory attributes (muscle color, flavor persistency, flavor intensity, bitterness, acidity, sweetness, saltiness, chewiness, juiciness, hardness, skatole odor, androsterone odor, odor intensity, fat distribution, muscle/fat, and fat color). The obtained results indicated that all dry-cured loin and “cachaço” were characterized by relatively high overall quality. For both dry-cured products, the different treatments applied to the animals’ feed had no significant effects on tasters’ evaluations of the attributes described above. Concerning dry-cured “cachaço”, flavor intensity (5.96–6.36), flavor persistency (5.78–6.05), muscle color (5.50–6.00), odor intensity (5.94–6.30), and juiciness (5.80–6.19) were the attributes with the highest scores by the panel of tasters for all the different treatments. As for the dry-cured loin, the most outstanding attributes were flavor intensity (5.78–6.05), flavor persistency (5.36–5.81), muscle/fat ratio (6.69–7.03), heterogeneous fat distribution (6.21–6.56), and odor intensity (5.76–6.01). On the other hand, the panel of tasters gave the lowest values for both dry-cured products of Bísaro pork for the attributes of acidity, bitterness, and rosterone and skatole odors. For the flavor intensity attribute, an average score of 5.93–6.36 for the dry-cured “cachaço” and 5.78–6.05 for the dry-cured loin was given. Lower values were also obtained by other authors for this attribute in dry-cured loin from different lines of Iberian pig (5.22–5.67) [4]. Regarding odor intensity, slightly lower values were obtained for the dry-cured loin of Entrepelado and Retinto lines for odor intensity [4]. The values obtained for the juiciness of the dry-cured loin in this study are in line with those obtained for the dry-cured loin of different lines of Iberian pigs with a commercial base diet [4]. The attributes of muscle/fat, hardness, and juiciness are directly related to animal diet. It is important to mention that the scores for flavor intensity for both products in this study were superior to those obtained by other authors for the dry-cured loin of Iberian pork [4]. The dry-cured “cachaço” had lower hardness scores and higher muscle/fat and juiciness scores compared to the dry-cured loin. According to Leite et al. [4], dry-cured “cachaço” has a higher total amount of fat, which is directly related to sensory assessments by panel tasters. This fact was also verified by the composition of the volatiles. The major aldehydes were significantly higher in the dry-cured “cachaço”, and these compounds were directly related to the amount of total fat. The sensory characteristics indicated that the use of different treatments in the animals’ feed resulted in higher acceptability of the final product. These results are in accordance with those reported by Fortin et al. [40] in which juiciness, tenderness, flavor, and absence of off-flavors were the most important attributes comprising the sensory experience during meat consumption.

In the present research, in order to minimize the differences between testers, the GPA was used to find a consensus. Figure 1b and Figure 2b show the biplot of the consensus configuration, with the correlations between the sensory attributes, treatments, and coordinates of the dry-cured “cachaço” and loin. For the dry-cured “cachaço” (Figure 1b), F1 and F2 together explained 62.49% of the total variability. Regarding the dry-cured loin (Figure 2b), F1 and F2 together explained 61.26% of the total variability. According to the coordinates of the different types of treatments and the correlation of the sensory attributes, panel tasters separated the treatments into three groups for both products, although they did not have the same trends. Concerning the dry-cured “cachaço”, the taster panel clearly separated treatments T1 (Basal diet and commercial feed) and T4 (Basal diet + 10% exhausted olive cake). Treatments T2 (Basal diet + 10% crude olive cake), T3 (Basal diet + 10% olive cake in two phases), and T5 (Basal diet + 10% exhausted olive cake + 1% olive oil) were grouped into one group. As far as the dry-cured “cachaço” is concerned, the tasters were able, through the 16 attributes studied, to group the T4 and T1 treatments separately. The other treatments were grouped together according to various sensory attributes. For the dry-cured loin, and across the 16 sensory attributes, treatments T4 and T5 were grouped together. This makes perfect sense since these treatments concerned the same type of olive cake except for the 1% fat added in T5. There was another group joining T2 and T1. Finally, T3 was separated from the other treatments.

## 4. Conclusions

The results obtained in the present study allow us to conclude that the introduction of olive industry byproducts, namely, olive cake, does not significantly influence two major acceptability parameters of dry-cured loin and “cachaço”: sensory analysis and volatile profile. Regarding sensory analysis, the panelists were not able to detect differences among the olive cakes introduced into the pig’s diets. This observation was directly related to the volatile profile, which was not significantly affected. The results show that the olive cake obtained by different methods of oil extraction does not negatively affect the sensory and volatile components of the processed meat products; thus, their appeal to the consumer is maintained.

## Figures and Tables

**Figure 1 foods-12-02499-f001:**
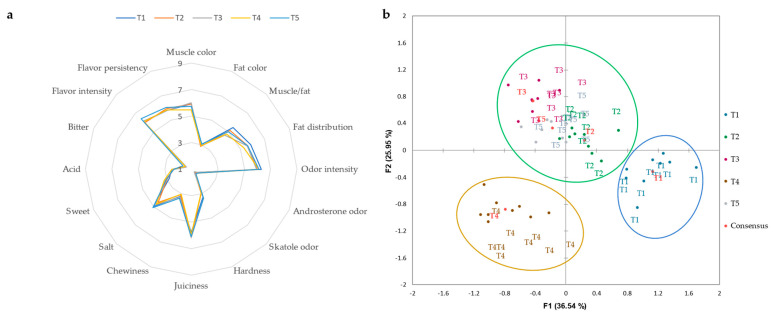
Sensory profile (**a**) and consensus configuration from GPA (**b**) of dry-cured “cachaço” of Bísaro pork.

**Figure 2 foods-12-02499-f002:**
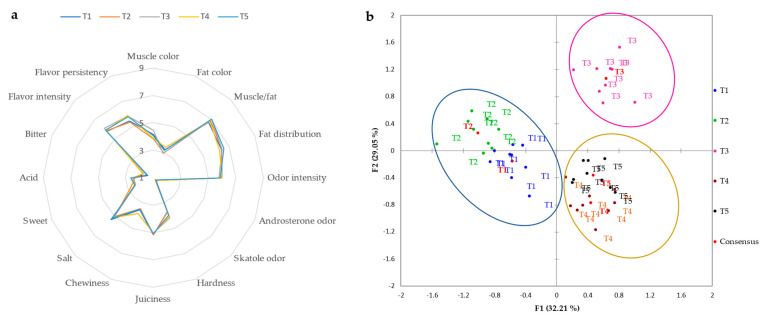
Sensory profile (**a**) and consensus configuration from GPA (**b**) of dry-cured loin of Bísaro pork.

**Table 1 foods-12-02499-t001:** Ingredient composition of the experimental diets (g/kg, as fed basis) and fatty acid composition (g/100 g).

			Diets		
	T1	T2	T3	T4	T5
Olive cake	0	10	10	10	10
Olive oil	0	0	0	0	1
Barley grain	45.80	41.20	41.20	41.20	41.20
Wheat grain	22.60	20.40	20.40	20.40	20.40
Soybean meal 47	12.90	11.60	11.60	11.60	11.60
Rice bran	5.00	4.50	4.50	4.50	4.50
Corn grain	2.50	2.20	2.20	2.20	2.20
DDG’s corn	5.00	4.50	4.50	4.50	4.50
Beet molasses	4.00	3.60	3.60	3.60	3.60
Minerals and vitamins	1.70	1.70	1.70	1.70	1.70
Supplement min+vit+fitase	0.50	0.30	0.30	0.30	0.30
Chemical composition of diet					
DM	98.05	98.49	98.19	98.15	98.46
OM	93.90	94.20	93.75	94.16	93.98
NDF	18.01	23.39	22.97	24.04	22.88
ADF	6.40	10.62	10.48	10.50	10.06
ADL	0.89	3.06	2.81	3.09	2.86
Cellulose	5.51	7.56	7.68	7.41	7.20
PB	16.00	13.38	13.45	14.39	13.98
GB	5.41	5.53	4.96	4.30	5.20
Fatty acids (g/100 g)					
C13:0	0.19	0.33	0.26	0.28	0
C14:0	0.18	0.13	0.14	0.18	0.14
C16:0	17.60	15.60	15.94	17.28	16.75
C16:1n-7	0.11	0.22	0.18	0.13	0.23
C18:0	2.06	2.54	2.32	2.09	2.88
C18:1n-9	25.85	39.93	36.05	27.37	33.79
C18:1n-7	0.99	1.32	1.17	0.96	1.17
C18:2n-6	47.55	35.31	39.06	46.34	40.76
C18:3n-3	2.64	2.05	2.28	2.64	2.33
C20:0	0.48	0.48	0.48	0.47	0.47
C20:1n-9	0.45	0.40	0.42	0.45	0.41
C22:0	0.29	0.27	0.27	0.29	0.27
C22:1n-9	0.48	0.37	0.37	0.43	0.39
C24:0	0.33	0.29	0.29	0.34	0.29
ΣSFA	21.54	20.03	20.09	21.31	20.57
ΣMUFA	28.00	42.36	38.31	29.45	36.10
ΣPUFA	50.46	37.61	41.60	49.24	43.34
PUFA/SFA	2.34	2.07	1.88	2.31	2.10
n-6/n-3	17.58	16.66	16.69	17.10	16.90

DM—dry matters; OM—organic matter; NDF—neutral detergent fiber; ADF—acid detergent fiber; ADL: acid detergent lignin; PB—crude protein; GB—crude fat. T1—Basal diet and commercial feed; T2—Basal diet + 10% crude olive cake; T3—Basal diet + 10% olive cake two phases; T4—Basal diet + 10% exhausted olive cake; T5—Basal diet + 10% exhausted olive cake + 1% olive oil.

**Table 2 foods-12-02499-t002:** Volatile compounds (expressed as AU 10^5^/g) of dry-cured loin and “cachaço”. Effect of treatment with olive cake on products and interactions between products and treatments.

Compounds Information			Dry-Cured Loin					Dry-Cured “Cachaço”					SEM	Sig. Product	Sig. Treat.
NAME	LRI	*m*/*z*	T1	T2	T3	T4	T5	T1	T2	T3	T4	T5			
Glycidol	512	43	6.64	9.40	9.97	11.40	8.60	5.91	6.94	6.45	6.76	6.49	0.586	0.025	0.648
2-Butanol, (R)-	610	59	0.56	0.36	0.93	0.52	12.99	0.25	1.01	0.17	0.36	0.83	1.203	0.292	0.343
1-Butanol	706	56	0.33	0.45	0.34	0.34	0.49	0.77	0.92	0.76	0.88	0.97	0.032	<0.001	0.308
2-Pentanol	749	45	2.59	2.13	1.58	1.99	2.24	1.20	0.73	1.00	0.94	1.01	0.136	<0.001	0.690
1-Butanol, 3-methyl-	805	55	12.36	16.23	13.67	14.78	10.69	3.12	3.11	2.68	4.79	2.65	1.133	<0.001	0.884
1-Pentanol	842	55	6.43	9.25	6.60	6.45	12.58	23.77	26.02	21.78	29.04	27.49	1.069	<0.001	0.429
2,3-Butanediol	911	45	41.95	20.62	30.50	38.45	19.30	37.27	51.67	41.90	47.41	20.78	3.934	0.211	0.392
1-Hexanol	947	56	12.66	25.09	17.64	18.68	29.29	8.16	11.00	9.35	9.18	10.70	1.489	<0.001	0.257
1-Octen-3-ol	1040	57	19.39	18.36	14.90	13.29	22.92	45.80	43.51	33.14	52.12	52.98	1.962	<0.001	0.273
Benzyl alcohol	1111	108	11.28	11.94	10.18	9.62	10.02	7.47	8.98	10.84	8.85	8.17	0.514	0.094	0.834
Terpinen-4-ol	1191	111	0.68	0.56	0.43	0.54	0.50	0.16	0.13	0.14	0.15	0.13	0.017	<0.001	0.140
Thymol	1290	150	0.09	0.09	0.07	0.12	0.08	0.02	0.02	0.01	0.01	0.02	0.004	<0.001	0.164
TOTAL ALCOHOLS			114.96	114.48	106.81	116.18	129.70	133.90	154.04	128.22	160.49	132.22			
Propanal	536	58	2.68	2.17	1.79	1.66	3.16	15.45	14.05	14.73	16.93	14.40	0.648	<0.001	0.960
Propanal, 2-methyl-	565	72	0.87	0.94	1.00	1.24	0.79	0.57	1.05	0.81	0.83	0.74	0.061	0.213	0.341
Butanal, 3-methyl-	660	58	7.75	8.84	11.65	12.75	7.37	2.65	5.05	4.69	4.71	3.45	0.875	0.003	0.595
Pentanal	727	58	9.01	8.21	6.10	6.46	11.51	24.34	25.60	21.57	27.44	26.35	0.881	<0.001	0.484
2-Butenal, 2-methyl-	798	84	1.04	1.16	1.22	1.87	1.26	0.51	1.29	0.90	0.92	1.16	0.153	0.275	0.737
Hexanal	842	55	162.54	153.55	140.13	130.40	183	291.52	320.50	287.24	325.08	302.33	9.878	<0.001	0.904
2-Hexenal, (E)-	931	83	0.19	0.20	0.17	0.15	0.26	1.34	0.97	0.96	1.30	1.37	0.050	<0.001	0.440
Heptanal	965	70	10.33	10.11	8.90	9.45	11.12	14.12	17.39	14.33	16.26	15.13	0.702	<0.001	0.887
Methional	990	104	0.37	0.40	0.50	0.52	0.38	0.05	0.17	0.14	0.08	0.10	0.021	<0.001	0.481
Furfural	993	96	0.02	0.02	0.02	0.01	0.05	0.18	0.15	0.16	0.21	0.17	0.008	<0.001	0.814
Benzaldehyde	1034	106	5.39	5.45	8.06	4.13	9.97	4.08	13.23	4.83	8.47	11.16	1.060	0.382	0.410
Octanal	1054	56	5.03	4.68	3.57	4.23	3.92	5.13	5.57	4.69	5.20	5.40	0.234	0.075	0.651
2,4-Heptadienal, (E,E)-	1067	81	0.05	0.03	0.03	0.03	0.06	1.67	1.23	1.33	2.07	1.77	0.088	<0.001	0.638
Benzeneacetaldehyde	1106	91	18.11	30.04	69.21	28.55	77.19	2.96	21.59	9.30	11.67	13.77	7.735	0.0391	0.615
2-Octenal, (E)-	1109	55	0.40	0.35	0.31	0.27	0.53	4.37	3.93	3.04	5.39	5.07	0.231	<0.001	0.548
Nonanal	1133	57	5.24	5.21	4.06	4.05	5.02	7.27	8.74	7.20	8.32	8.06	0.281	<0.001	0.653
2-Nonenal, (E)-	1184	70	0.47	0.35	0.29	0.35	0.38	0.72	0.76	0.68	0.82	0.66	0.035	<0.001	0.883
2-Decenal, (E)-	1253	70	0.20	0.21	0.15	0.17	0.18	0.30	0.29	0.25	0.30	0.27	0.014	<0.001	0.719
TOTAL ALDEHYDES			229.69	231.92	257.16	206.29	316.15	377.23	441.56	376.85	436.00	411.36			
NAME	LRI	*m*/*z*	T1	T2	T3	T4	T5	T1	T2	T3	T4	T5			
Acetic acid	689	60	0.30	0.19	0.17	0.23	0.20	0.49	0.31	0.40	0.34	0.23	0.018	<0.001	0.248
Butanoic acid	910	60	2.60	2.06	3.33	3.86	3.51	21.59	15.85	22.42	19.87	13.90	0.834	<0.001	0.358
Butanoic acid, 3-methyl-	959	60	5.65	1.97	2.93	6.68	1.93	5.56	4.03	1.49	5.50	1.51	0.828	0.899	0.337
Hexanoic acid	1071	60	0.95	0.88	0.60	0.92	1.46	15.45	7.79	7.81	14.73	16.75	1.224	<0.001	0.617
TOTAL ACIDS			9.50	5.10	7.03	11.69	7.10	43.09	27.98	32.12	40.44	32.39			
Pentane	527	57	0.66	1.37	0.67	0.79	1.48	3.15	2.25	2.49	2.31	2.14	0.152	<0.001	0.874
1,4-Pentadiene	539	67	1.05	2.28	1.04	2.05	2.37	1.87	1.21	1.77	2.57	1.16	0.242	0.981	0.734
Cyclopentane, 1,2-dimethyl	665	56	0.29	0.31	0.16	0.20	0.26	0.28	0.26	0.25	0.26	0.24	0.020	0.649	0.666
Heptane	673	71	10.90	17.59	11.79	13.99	13.58	10.19	9.17	10.51	6.98	8.02	0.757	0.003	0.697
Octane	817	85	36.06	49.53	35.05	42.34	43.57	29.16	28.19	28.87	21.62	23.48	1.932	<0.001	0.735
2-Octene, (E)-	828	55	0.17	0.20	0.15	0.16	0.29	0.18	0.17	0.18	0.19	0.22	0.010	0.934	0.051
Nonane	927	57	0.88	0.89	0.64	0.79	0.58	0.61	0.46	0.55	0.43	0.72	0.039	0.010	0.758
α-Phellandrene	961	93	4.90	4.16	3.36	5.39	3.58	1.78	1.26	1.45	1.72	1.29	0.202	<0.001	0.219
Hexane, 3,3-dimethyl	927	85	0.48	0.46	0.35	0.44	0.35	0.32	0.25	0.28	0.24	0.37	0.020	0.003	0.696
Butane, 2,2,3-trimethyl-	994	57	0.39	0.35	0.21	0.32	0.21	0.16	0.15	0.15	0.14	0.16	0.016	<0.001	0.306
Nonane, 5-methylene-	999	56	0.48	0.47	0.28	0.37	0.23	0.18	0.17	0.18	0.16	0.18	0.022	<0.001	0.332
Decane	1018	57	2.13	3.68	1.89	2.03	1.52	1.02	0.69	0.51	0.80	0.97	0.153	<0.001	0.248
β-Myrcene	1021	93	1.16	1.09	0.73	1.30	0.73	0.35	0.24	0.26	0.38	0.27	0.054	<0.001	0.166
(Z)-4-Methyl-2-hexene	1043	98	0.83	0.75	0.54	0.68	0.44	0.26	0.30	0.28	0.27	0.29	0.036	<0.001	0.488
2,2,4,4-Tetramethyloctane	1048	57	33.59	31.00	23.97	30.09	19.66	14.07	15.60	15.46	13.73	14.59	1.376	<0.001	0.526
γ-Terpinene	1078	93	3.55	2.66	2.41	3.91	2.33	0.89	0.69	0.80	1.00	0.77	0.142	<0.001	0.153
Undecane	1098	85	0.51	0.49	0.39	0.47	0.34	0.28	0.26	0.25	0.24	0.29	0.019	<0.001	0.613
Dodecane, 2,6,10-trimethyl-	1171	57	0.61	0.57	0.47	0.58	0.48	0.32	0.34	0.30	0.30	0.38	0.027	0.001	0.914
Dodecane	1171	71	0.46	0.45	0.34	0.42	0.35	0.25	0.25	0.20	0.24	0.26	0.019	<0.001	0.594
2-Heptene, 3-methyl	1184	83	0.34	0.30	0.24	0.27	0.30	0.58	0.55	0.51	0.72	0.60	0.031	<0.001	0.886
Cyclododecane	1206	83	0.57	0.53	0.42	0.53	0.43	0.29	0.32	0.25	0.29	0.32	0.023	<0.001	0.668
TOTAL HYDROCARBONS			100.01	119.13	85.10	107.12	93.08	66.19	62.78	65.50	54.59	56.72			
NAME	LRI	*m*/*z*	T1	T2	T3	T4	T5	T1	T2	T3	T4	T5			
2,3-Butanedione	594	86	0.76	0.25	0.32	0.61	0.30	0.92	1.39	1.15	0.90	0.54	0.082	0.002	0.470
2-Butanone	599	72	1.48	2.04	2.92	2.84	18.00	1.66	4.43	3.35	2.57	2.79	1.536	0.413	0.379
1-Penten-3-one	715	55	0.06	0.04	0.04	0.04	0.08	0.89	0.61	0.62	0.80	0.97	0.037	<0.001	0.334
2-Pentanone	719	86	5.51	5.17	3.41	5.00	4.07	4.94	6.67	4.98	3.61	3.31	0.629	0.918	0.810
2,3-Pentanedione	735	57	4.42	3.04	3.37	14.81	9.85	5.97	6.41	7.58	6.25	6.25	1.412	0.848	0.651
Acetoin	786	45	19.38	8.32	10.23	24.94	8.93	17.72	27.18	22.85	14.72	6.06	2.058	0.424	0.345
2-Hexanone	854	58	1.60	1.94	1.16	1.51	1.15	1.02	1.17	0.91	0.73	0.79	0.136	0.052	0.658
2-Heptanone	959	58	14.02	18.12	12.61	14.78	13.59	17.53	17.70	9.86	15.38	16.08	1.335	0.784	0.620
4-Hexen-3-one, 5-methyl-	1031	83	0.28	0.38	0.30	0.36	0.35	0.30	0.30	0.27	0.30	0.41	0.024	0.513	0.675
Butyrolactone	1034	86	0.77	0.64	0.81	1.07	0.71	1.89	1.76	1.72	1.96	1.67	0.048	<0.001	0.334
3,5-Octadien-2-one	1121	95	0.72	0.41	0.46	0.46	0.83	2.05	1.73	1.78	2.62	2.37	0.141	<0.001	0.662
2-Nonanone	1126	58	2.07	2.29	2.00	2.23	1.69	2.66	2.98	0.87	2.60	2.46	0.284	0.654	0.710
TOTAL KETONES			51.07	42.64	37.63	68.65	59.55	57.55	72.33	55.94	52.44	43.70			
Acetic acid, methyl ester	547	74	0.30	0.19	0.17	0.23	0.20	0.49	0.31	0.40	0.34	0.23	0.025	0.008	0.188
Ethyl Acetate	603	61	10.70	9.71	8.21	9.11	10.78	5.10	3.91	2.26	4.00	4.53	0.578	<0.001	0.605
Propanoic acid, ethyl ester	735	102	0.52	0.28	0.30	2.81	1.63	0.23	0.15	0.11	0.27	0.22	0.291	0.123	0.555
n-Propyl acetate	742	61	0.15	0.18	0.17	0.25	0.62	0.08	0.13	0.06	0.09	0.09	0.042	0.029	0.317
Propanoic acid, 2-methyl-, ethyl ester	798	71	3.36	2.65	1.92	2.55	1.91	0.90	0.60	0.41	0.93	0.61	0.236	<0.001	0.698
Butanoic acid, ethyl ester	850	71	10.68	11.53	9.61	11.64	11.15	14.04	8.08	8.05	12.13	12.72	1.017	0.956	0.753
Butanoic acid, 2-methyl-, ethyl ester	902	102	7.83	5.11	3.49	5.09	3.41	1.72	1.57	0.94	2.00	1.33	0.578	0.004	0.632
Butanoic acid, 3-methyl-, ethyl ester	906	88	15.81	9.00	7.01	9.89	6.78	3.08	3.53	1.93	3.49	2.31	1.147	0.004	0.642
1-Butanol, 3-methyl-, acetate	932	70	0.63	0.43	0.41	0.41	0.93	0.65	0.50	0.43	0.70	0.59	0.049	0.938	0.191
1-Butanol, 2-methyl-, acetate	935	70	0.25	0.08	0.09	0.11	0.21	0.03	0.06	0.07	0.03	0.02	0.014	<0.001	0.508
Hexanoic acid, ethyl ester	1039	88	8.15	11.98	7.60	10.20	12.60	8.57	7.47	6.38	9.59	12.74	0.848	0.372	0.294
Octanoic acid, ethyl ester	1187	88	3.38	4.79	3.08	3.53	5.54	3.39	3.19	1.95	3.93	4.41	0.348	0.294	0.262
Decanoic acid, ethyl ester	1314	88	2.16	3.04	1.87	1.98	3.12	2.14	1.75	0.84	2.31	2.20	0.231	0.197	0.465
TOTAL ESTERS			63.92	58.97	43.93	57.80	58.88	40.42	31.25	23.83	39.81	42			
NAME	LRI	*m*/*z*	T1	T2	T3	T4	T5	T1	T2	T3	T4	T5			
Furan, 2-ethyl	702	81	0.86	0.80	0.33	0.70	0.82	1.50	1.34	1.28	1.60	1.22	0.082	<0.001	0.640
2-n-Butyl furan	935	81	0.85	0.74	0.53	0.66	0.80	1.70	1.33	1.05	1.78	1.40	0.086	<0.001	0.411
Furan, 2-pentyl	1026	81	8.27	7.25	5.11	7.48	7.92	10.00	9.30	6.80	10.64	9.49	0.781	0.207	0.696
Furan, 2-propyl	1078	81	0.17	0.15	0.14	0.14	0.26	1.83	1.72	1.62	2.46	2.49	0.098	<0.001	0.410
TOTAL FURAN			10.15	8.94	6.11	8.98	9.80	15.03	13.69	10.75	16.48	14.60			
Sulfide, allyl methyl	699	88	8.03	4.91	7.04	11.53	3.39	6.27	4.52	5.40	3.38	3.70	0.967	0.253	0.644
1H-Pyrrole, 3-methyl	792	81	0.13	0.14	0.09	0.13	0.12	0.17	0.16	0.21	0.13	0.10	0.009	0.089	0.555
1,3-Benzenediol, monobenzoate	829	105	0.85	1.18	0.96	1.03	1.06	0.48	0.34	0.29	0.53	0.58	0.055	<0.001	0.780
Pyrazine, 2,6-dimethyl-	969	108	2.64	2.51	2.23	2.64	1.80	2.66	1.65	2.37	2.02	1.42	0.227	0.486	0.664
OTHERS COMPOUND			11.65	8.74	10.32	15.33	6.37	9.58	6.67	8.27	6.06	5.80			

Sig.: significance; ns—not significant; SEM—standard error of the mean; LRI: linear retention index calculated for DB-624 capillary column installed on a gas chromatograph equipped with a mass selective detector; *m*/*z*: quantifier ion. Treatments: T1—Basal diet and commercial feed; T2—Basal diet + 10% crude olive cake; T3—Basal diet + 10% olive cake in two phases; T4—Basal diet + 10% exhausted olive cake; T5—Basal diet + 10% exhausted olive cake + 1% olive oil.

## Data Availability

All data were presented in the manuscript. Data can be requested from the corresponding author via email.

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
