# Peer review of "Did the Addition of Olive Cakes Obtained by Different Methods of Oil Extraction in the Finishing Diet of Bísaro Pigs Affect the Volatile Compounds and Sensory Characteristics of Dry-Cured Loin and “Cachaço”?"

_foods, 2023, doi:10.3390/foods12132499_

Round 1

Reviewer 1 Report

The present study investigated the effect of  the inclusion of olive cake in the finishing diets of  Bísaro pigs on the volatile compounds and sensory characteristics 3 of dry-cured loin and dry-cured “cachaço”. The study was designed carefully and carried out properly. Authors showed that inclusion of olive cake in diets  did not affect acceptability parameters of dry-cured loin and dry-cured “cachaço in terms of volitile profile and sensory score. It may contribute to the deveolping  of a new application of olive oil byproducts to reduce the amount of this olive waste to process and keep friendly to the enviroment. However, there are a few questions should be clarified:1 as mentioned by authors that olive cake is hazardous and toxic by-product, whether long-term olive cake intake will damage the health of pigs and depress fed intake and growth? authors should provide the necessary data or information to clarify it. 2. Why did the study designed two kinds of olive cakes inclduing the  olive cake and exhauted olive cake  even added 1% olive oil? Beside the different contents of moisture and oil, is there other components varying?

Author Response

Dear review,

Thank you for your comments. We hope that our answers have been enlightening.

Thank you for your attention.

REV.1

The present study investigated the effect of the inclusion of olive cake in the finishing diets of Bísaro pigs on the volatile compounds and sensory characteristics 3 of dry-cured loin and dry-cured “cachaço”. The study was designed carefully and carried out properly. Authors showed that inclusion of olive cake in diets did not affect acceptability parameters of dry-cured loin and dry-cured “cachaço in terms of volatile profile and sensory score. It may contribute to the developing of a new application of olive oil byproducts to reduce the amount of this olive waste to process and keep friendly to the environment.

Response: first, our thanks for the praise of our work. This study is part of a project that was very well thought out and that can be applied to farms and the processing industry of meat from the Bisaro breed. It is necessary to create projects and consequently publications that make a difference in the food sector and beyond. It is necessary to create strategies that have real applicability and that get off the paper.

However, there are a few questions should be clarified:1 as mentioned by authors that olive cake is hazardous and toxic by-product, whether long-term olive cake intake will damage the health of pigs and depress fed intake and growth? authors should provide the necessary data or information to clarify it.

Response: This project has several actors, and the initial part of this work aimed to study what percentages of olive cake the Bísara breed pigs could ingest without harming their growth and digestibility. In the work already done by the UTAD researchers (which is still being published) they do not indicate any problem in the digestibility of the animals nor that there is any indication of a long-term problem. We quote below some of the articles already published by these researchers, to prove their experience in this type of issues:

            Effect of the dietary incorporation of olive leaves on growth performance, digestibility, blood parameters and meat quality of growing pigs.

            Effect of the dietary incorporation of untreated and white-rot fungi (Ganoderma resinaceum Bound) pre-treated olive leaves on growing rabbits.

            The effect of olive leaves supplementation on the feed digestibility, growth performances of pigs and quality of pork meat.

            The effect of olive leaf supplementation on the constituents of blood and oxidative stability of red blood cells.

Why did the study designed two kinds of olive cakes including the olive cake and exhausted olive cake even added 1% olive oil? Beside the different contents of moisture and oil, is there other components varying?

Response: The T4 treatment (with extracted olive cake) was chosen to create a new T5 treatment but with a small percentage of olive oil (1%). The choice of this olive cake was due to the fact that this olive cake has a more limited composition in fat, and with this inclusion (1% olive oil) we could give an added value that could be reflected in the quality of the meat and its processed products. It should also be noted that this type of olive cake (extracted) is easier to transport, has greater conservation capacity and is available for a longer period of time.

Reviewer 2 Report

Dear authors,

Thank you for your great effort, the comment after peer-reviewing process can be obtained in the attached document.

Thank you,

Sincerely

The English proofread are highly suggested for this manuscript.

Author Response

Dear review,

All modifications were made following the reviewer's suggestions and comments, and responses to their comments are also attached. Thanks to their recommendations, significant modifications were made throughout the manuscript.

Thank you for your attention.

REV.2

Abstract:

L23                   : Please mention all treatment details or delete the phrase five treatments.

Response: Suggestion accepted; changes have been made in the revised version of the manuscript.

L23-L25            : It is better to mention detail of treatments rather than mentioning different in water/fat content.

Response: Suggestion accepted; changes have been made in the revised version of the manuscript.

L25                   : were based.

Response: Suggestion accepted; changes have been made in the revised version of the manuscript.

Abstract should also contain the theory and practical importance, please provide conclusion on what treatment should be fed to Bisaro pigs according to the study.

Response: Suggestion accepted; changes have been made in the revised version of the manuscript.

Introduction

L44                   : near extension due to the modern society’s eating ~~~

Response: near extinction and not extension

Please write Trás-os-Montes in italic.

Response: Suggestion accepted; changes have been made in the revised version of the manuscript.

L51-L54 contains writing error, please revise it accordingly.

Response: Suggestion accepted; changes have been made in the revised version of the manuscript.

L63                   : Performance  functions / similar meaning.

Response: Suggestion accepted; changes have been made in the revised version of the manuscript.

L71-L73            : Please elaborate in detail what were the conclusion of each of previous study and the research gap with the current study by authors.

Response: Suggestion accepted; changes have been made in the revised version of the manuscript.

This article contains very few information related to the olive cake and cachao, whereas these are very essential factors to be explained clearly within the introduction, therefore please revise manuscript accordingly with the addition of necessary information.

Response: our research group has previously authored of other papers on this topic, and we have extensively covered information related to olive cake, which may lead to issues with repeaatibility percentages. Moreover, we strongly believe that our colleagues at UTAD possess the necessary expertise and precision to address this subject in a manner that aligns with the reviewer´s expectations regarding rigor and comprehensive description. Publications about the influence of different oilcakes on the digestibility of animals by our colleagues are about to be sent for possible publication

Materials and Methods

Please mention the average age and total number of the animal.

Response: Suggestion accepted; changes have been made in the revised version of the manuscript.

L93                   : What is the meaning of basic diet and commercial feed?

Response: These animals are in a semi-extensive regime, with access to horticultural products from the region (turnips, potatoes, cabbage, fruit, etc.) plus the typical commercial pig diet.

  • L93-L97 is not clear for me on how the authors came up with this design? Why did the olive oil only be added at only one group of treatments?

Response: The initial part of the project where this work is included was aimed at testing the various types of olive cake with different percentages, with 10% being the value considered adequate taking into account several factors. Regarding the T4 and T5 treatments, the olive cake is the same but T5 has an addition of 1% olive oil. This inclusion of olive oil is due to the fact that this type of olive cake has a more limited fat composition.

  • The fatty acid profile of the diets used were made in Meat Technology Center of Galicia, Ourense – Spain?

Response: Yes

  • Materials and methods did not present anything but the explanation on how the experiment was conducted. Meanwhile, in this study, authors tended to mention the result trends of the study. Please revise it accordingly to meet the requirement given.

The research design of this experiment is fail to accommodate the effect of diets on different style of dry-cured loin.

Response: The material and methods aim to explain the procedures carried out in the study performed. We believe that we have explained all the steps succinctly, including the processing of the loins and shoulders. These types of procedures carried out in a factory unit are sometimes poorly explained due to the secrecy of the industrial units, which did not happen in this work. Therefore, we were not able to understand the specific issue. Please try to be clearer so that we can understand.

Results and Discussions

  • Please rephrase the paragraph 1 of L182-L189 because it is potentially confusing for readers.

Response: Suggestion accepted; changes have been made in the revised version of the manuscript.

  • It is highly expected that the authors could elaborate in such a consistent way: i. e.:
  • Mention the result trend / data obtained from the experiment. Please clearly present the detail observation.
  • Compare with the other study/related experiments with regard to the obtained data.
  • Mention the possible mechanism of how the result could be obtained.

Response: While we acknowledge the need for improving the discussion of our results, we kindly request a more specific and detailed revision from you. We believe that the current review lacks clarity and that any modifications or enhancements might not align with your requirements. Nevertheless, we are confident that we have provided a thorough explanation of our results, incorporating comparisons with other studies and establishing connections between our findings and potential influencing mechanisms.

Conclusion

The conclusion is sufficient, please also mention the summary of conclusion within the abstract.

Response: Suggestion accepted; changes have been made in the revised version of the manuscript.

Reviewer 3 Report

1.      Dear Authors, the manuscript is interesting but I suggest to Editor to reject it in the current  form. The manuscript contains many errors and omissions. Detailed analysis below.

      Title should be changed – not only olive cakes were in diets but also olive oil, so it is not precisive.

2.      Abstract

L 23-24 - different types of olive cakes (five treatments) – it suggests 5 different cakes types– it is not true.

Lack of information about differences among groups and about experiment, number of pigs and samples.

L25-26, and 33 – there is not data about water and fat content in meat of these pigs in the manuscript.

Lack of conclusion.

3.      Introduction

L. 43-45 - There were several reasons why there was a decline of the breed, leading to its near extinction (modern society´s eating habits, opting for lean meats; rural exodus, etc.) – the most important description in this sentence is in the scopes – remove scopes and rewrite sentence. This same in line 55-57.

L 51-53 – if these products are not known describe it briefly and give some details about differences between them.

L 63-65 – these are spices and you must concern on olive cakes – what is special in them? What do you expect and why? What kind of volatile substances were important? What is mechanism of production of volatile substances from diet?

L 70-74 – it is repetition of L 51-57.

L 82-86 – the aim should be simple. This description should be in M&M part.

What`s about welfare of pigs? Any standards?

L 97 - In total, forty animals of the Bísaro breed were used. – not necessary – 5 x 8=40

L 98-99 – ad libitum - in italic

L 100-107 – it should be in table

L. 108-109 – where are these results?

.

4.      Materials and Methods

 M&M section should be carefully sorted! The section is poorly informative.

There are no tables with diets and their nutritional values. There is also no basic information about olives cakes and their value.

L. 93 what is “Basic diet and commercial feed” ???

L 94 what is “Basic diet + 10% olive cake in two phases” - you don`t meaning about any phases in manuscript.

What is the difference between exhausted olive cake and olive cake? What is the characteristic of olive oil?

What mass had the pigs at the start and at the finish? Was one or more phases??

L 119-125 – differences should be described in introduction.

L 130 and others - ºC should be removed by ℃

Were the differences in meat composition among groups?

Is the taste of products connected with fat content? Were the differences in this trial?

Statistic – I don`t understand why you compare both products together if they are different. They should be analyzed separately.

5.      Tables

What is LRI in tables??? What is m/z? Groups and short names should be described under the Table 1.

The titles of tables are not informative and too long – it should be added which groups of substances are in tables.

6.      Results and discussion

L 393 – what means „???”

„Table” you write with big letter “figure” with low letter – be consistent.

Is the correlation or similarity between sensual and chemical analysis?

L 420-430 – what is the conclusion from that?

7.      Conclusion

L 435 – „volatile profile” – is it correct?

Can you identify „sensory analysis and volatile profile as two parameters”?

L 436 – pigs` diets

L 438-443 - there is not conclusion from this study – generally no effect of olive cakes was found.

8.                          Literature is not prepared in accordance with the requirements.

Author Response

Dear review,

All modifications were made following the reviewer's suggestions and comments, and responses to their comments are also attached. Thanks to their recommendations, significant modifications were made throughout the manuscript.

Thank you for your attention.

REV.3

Dear Authors, the manuscript is interesting, but I suggest to Editor to reject it in the current form. The manuscript contains many errors and omissions. Detailed analysis below.

Title should be changed – not only olive cakes were in diets but also olive oil, so it is not precisive.

Response: we do not think it is appropriate to change the title of the article. Adding 1% olive oil would only create entropy in the article, adding nothing new to the reader.

2.Abstract

L 23-24 - different types of olive cakes (five treatments) – it suggests 5 different cakes types– it is not true.

Response: Suggestion accepted; changes have been made in the revised version of the manuscript.

Lack of information about differences among groups and about experiment, number of pigs and samples.

Response: Suggestion accepted; changes have been made in the revised version of the manuscript.

L25-26, and 33 – there is not data about water and fat content in meat of these pigs in the manuscript.

Response: To avoid including bibliographical references in the abstract, which is uncommon in our scientific field, we have instead referred to the significant parameters obtained from previous studies involving the animals in our research. However, it is important to note that the references have been appropriately included throughout the entirety of the paper.

Lack of conclusion.

Response: Suggestion accepted; changes have been made in the revised version of the manuscript.

3.Introduction

  1. 43-45 - There were several reasons why there was a decline of the breed, leading to its near extinction (modern society´s eating habits, opting for lean meats; rural exodus, etc.) – the most important description in this sentence is in the scopes – remove scopes and rewrite sentence. This same in line 55-57.

Response: Suggestion accepted; changes have been made in the revised version of the manuscript.

L 51-53 – if these products are not known describe it briefly and give some details about differences between them.

Response: we consider that the description we give in the introduction and further explanation in the material and methods is what the reader needs.

L 63-65 – these are spices and you must concern on olive cakes – what is special in them? What do you expect and why? What kind of volatile substances were important? What is mechanism of production of volatile substances from diet?

Response: Spices are extremely important in volatile compounds and are key in the development of terpenes and sulfide compounds. These compounds impart flavor to cured products, which makes them more acceptable to consumers. The volatile compounds derived from the oxidation of unsaturated fatty acids, namely through the lipoxygenase pathway, Maillard reactions and the Streaker degradation of some amino acids.

L 70-74 – it is repetition of L 51-57.

Response: we do not consider this information to be a repetition. We think it should be kept

L 82-86 – the aim should be simple. This description should be in M&M part.

Response: we think that the objective is simple, we just think it is important to mention the name of the treatments, which makes the sentence longer, but this does not mean that the objective is not simple.

AIM - “So, the aim of this study was to evaluate the effect of the inclusion of different types of olive cake in the finishing diet on volatile compounds of dry-cured loin and dry-cured “cachaço” from Bísaro pig”

What`s about welfare of pigs? Any standards?

Response: The Bísaro pigs were slaughtered at the Municipal Slaughterhouse of Bragança as described by Álvarez-Rodríguez and Teixeira and in accordance with applicable legislation of welfare of pigs.

L 97 - In total, forty animals of the Bísaro breed were used. – not necessary – 5 x 8=40

Response: Eight animals were used for each treatment (control, T1, T2, T3, T4 and T5), making a total of 40 animals. We don't understand the question.

L 98-99 – ad libitum - in italic

Response: Suggestion accepted; changes have been made in the revised version of the manuscript.

L 100-107 – it should be in table

Response: We also consider that presenting this information in a table format would be beneficial. However, we have encountered challenges with the repeatability percentage in the past. In previous articles, we included this information in a table; however, for this particular instance, we intend to provide a concise summary along with a bibliographic reference.

  1. 108-109 – where are these results?

Response: table 1 in the article: Leite, A.; Domínguez, R.; Vasconcelos, L.; Ferreira, I.; Pereira, E.; Pinheiro, V.; Outor-Monteiro, D.; Rodrigues, S.; Lorenzo, J.M.: Santos, E.M.; Andrés, S.C.; Campagnol, P, C, B.; Teixeira, A. Can the introduction of different olive cakes affect the carcass meat and fat quality of Bísaro pork? Foods, 2022, 11, 1165. https://doi.org/10.3390/foods11111650

  1. Materials and Methods

M&M section should be carefully sorted! The section is poorly informative.

There are no tables with diets and their nutritional values. There is also no basic information about olives cakes and their value.

Response two observation: We disagree with this observation. In fact, we believe that our work contains information that many published articles do not include, such as the manufacturing flowchart of the processed products. Typically, studies related to the industry are limited in terms of providing information about their manufacturing steps. Moreover, we have even included previously published articles with the same animals to provide additional information. Since the materials and methods are practically identical, we may be subject to scrutiny for high repeatability, as has happened before.

  1. 93 what is “Basic diet and commercial feed” ???

Response: These animals are in a semi-extensive regime, with access to horticultural products from the region (turnips, potatoes, cabbage, fruit, etc.) plus the typical commercial pig diet.

L 94 what is “Basic diet + 10% olive cake in two phases” - you don`t meaning about any phases in manuscript.

Response: Centrifuged olive cake can exhibit either a biphasic or three-phase nature. It is only the centrifuged olive cake that can possess two or three phases, thus justifying our description in this manner. Once again, articles from our project partners are currently being published, offering clear discussions on the different types of olive cake and providing comprehensive descriptions on the subject.The two-phase system is more ecofriendly, as the olive cake is moister and contains lower oil content, due to the more efficient separation of the oil from the other elements by the centrifugation system. The three-phase olive cake is drier, as during the extraction of olive oil, the olive paste is separated into three parts: olive oil, dry olive cake and olive mill wastewaters. The objective and expertise of our research group do not aim to provide information about olive cake to the reader, but rather to investigate its impacts on processed products and whether its inclusion may lead to rejection by end consumers.

What is the difference between exhausted olive cake and olive cake? What is the characteristic of olive oil?

Response: The olive cakes can also be characterized by their composition and oil content, as crude olive cake and extracted olive cake. The exhausted olive cake is easier to transport, has greater conservation capacity and is available for a longer period of time that olive cake. Regarding the added olive oil, it does not pertain to a single variety of olives, but rather the oil obtained from the processing units in the Trás-os-Montes region

What mass had the pigs at the start and at the finish? Was one or more phases??

Response: The various treatments were administered to the animals (in batches) during the finishing phase, specifically in the last 90 days prior to their slaughter. Data collection took place daily to assess weight gains and other relevant information. However, once again, these data belong to our project partners, whose publication of the studies is currently in the finalization phase.

L 119-125 – differences should be described in introduction.

Response: We do not believe that this information should be included in the introduction in a detailed manner. Instead, it should be presented in the materials and methods section.

L 130 and others - ºC should be removed by â„ƒ

Response: Suggestion accepted; changes have been made in the revised version of the manuscript.

Were the differences in meat composition among groups?

Response: there were differences in the chemical composition of the meat. These differences are described in other works that serve as a bibliography for the current work. The works follow:

Leite, A.; Domínguez, R.; Vasconcelos, L.; Ferreira, I.; Pereira, E.; Pinheiro, V.; Outor-Monteiro, D.; Rodrigues, S.; Lorenzo, J.M.: Santos, E.M.; Andrés, S.C.; Campagnol, P, C, B.; Teixeira, A. Can the introduction of different olive cakes affect the carcass meat and fat quality of Bísaro pork? Foods, 2022, 11, 1165. https://doi.org/10.3390/foods11111650

Leite, A.; Vasconcelos, L.; Ferreira, I.; Domínguez, R.; Pereira, E.; Rodrigues, S.; Lorenzo, J. M.; Teixeira, A. Effect of the inclusion of olive cake in the diet on the physicochemical characteristics of dry-cured loin and dry-cured “cachaço” of Bísaro pig. Applied Sci. 2023, 13, 1439. https://doi.org/10.3390/app13031439

Is the taste of products connected with fat content? Were the differences in this trial?

Response: the amount of fat in both products is very different, being significantly higher in the steaks. The amount of fat present in a product interferes with the evaluation of taste, texture, flavor, chewiness, and other sensory parameters. The issue of fat is referred to throughout the document.

Statistic – I don`t understand why you compare both products together if they are different. They should be analyzed separately.

Response: we were unable to perceive this issue. Both products were evaluated individually, statistically speaking the statistical differences between them were evaluated. The goal of this work was to try to understand if both products were different in terms of volatile composition, and already published articles show that both products are very different in terms of their chemical composition.

5.Tables

What is LRI in tables??? What is m/z? Groups and short names should be described under the Table 1.

Response: LRI: linear retention index calculated for DB-624 capillary column installed on a gas chromatograph equipped with a mass selective detector; m/z: quantifier ion. Information included in table 1

The titles of tables are not informative and too long – it should be added which groups of substances are in tables.

Response: we don't consider that the information replicated in the table title adds information to the reader. We think this way is much easier and more effective.

6.Results and discussion

L 393 – what means „???”

Response: A greater intensity of flavor gives greater experience of stronger and distinctive flavor. Suggests a more pronounced flavor profile.

„Table” you write with big letter “figure” with low letter – be consistent.

Response: Suggestion accepted; changes have been made in the revised version of the manuscript.

Is the correlation or similarity between sensual and chemical analysis?

Response: The sensory evaluation by the tasters is related to the amount of fat in each of the products.

L 420-430 – what is the conclusion from that?

Response: The purpose was to understand if this analysis, taking into account the evaluation by the panel of tasters, could group or not the different treatments. Based on the 16 sensory parameters evaluated, this discriminant was able to separate part of the treatments, that is, there are characteristics that separate them from each other.

7.Conclusion

L 435 – „volatile profile” – is it correct?

Response: yes, is correct

Can you identify „sensory analysis and volatile profile as two parameters”?

Response: Yes, but we don't understand the point of the question. These are the two issues addressed in the paper

L 436 – pigs` diets

Response: Suggestion accepted; changes have been made in the revised version of the manuscript.

L 438-443 - there is not conclusion from this study – generally no effect of olive cakes was found.

Response: This statement does not seem correct to us at all. Not all studies have to show significant differences to be validated and useful. In our case it is really positive that there are no significant differences in the volatiles profile and the evaluation by the tasters. This shows that the product is not negatively affected by the inclusion of olive pomace in the animals' diet. And that it could therefore be a circular economy option. A product considered very toxic can be part of the animal feed of pigs of the Bisaro breed, without adversely affecting its composition.

8.Literature is not prepared in accordance with the requirements.

Response: the bibliography used is based on other authors who have studied volatile compounds in processed products from other pig breeds, the use of by-products in the industry, the sensory evaluation of tasters, etc. We consider that the bibliography used is fully adjusted to our work.

Reviewer 4 Report

The authors failed to persuade the readers of the importance of studying olive cake, which could influence the products (dry-cured loin and dry-cured "cachaço"). The authors were required to evaluate previous publications and then explain how adding olive cake in pig diets impacted the products stated.

This was most likely a factorial design. There were significant product x treatment interactions based on SEM of each parameter in all tables. Please check again.

The authors did not highlight different varieties of olive cake added to the diets that contributed to the assessed parameters in the results and discussion.

Finally, please be concise and explicit about what you discovered in this work. Because your data showed that treatments had no meaningful impacts, no additional comments were made to the conclusion.

Author Response

Dear review,

Thank you for your comments. We hope that our answers have been enlightening.

Thank you for your attention.

REV 4

The authors failed to persuade the readers of the importance of studying olive cake, which could influence the products (dry-cured loin and dry-cured "cachaço"). The authors were required to evaluate previous publications and then explain how adding olive cake in pig diets impacted the products stated.

Response: first of all, we thank you for your opinion about the work. In our opinion, the study that we have been carrying out for more than two years is extremely important. It is a work that began in a pig farm with Bisara breed animals, which were accompanied by researchers from the University of Trás-os-Montes e Alto Douro (UTAD). They evaluated what percentages of olive cake could be applied to the animals, considering several factors (digestibility, toxicity). The results (which have not yet been published) revealed a very similar live weight gain and average feed intake between the various treatments. Furthermore, no negative data was observed, making clear the possibility of the use and valorization of various olive cake in the feeding of Bisaro pigs. We quote below some of the articles already published by these researchers, to prove their experience in this type of issues:

Effect of the dietary incorporation of olive leaves on growth performance, digestibility, blood parameters and meat quality of growing pigs.

            Effect of the dietary incorporation of untreated and white-rot fungi (Ganoderma resinaceum Bound) pre-treated olive leaves on growing rabbits.

            The effect of olive leaves supplementation on the feed digestibility, growth performances of pigs and quality of pork meat.

            The effect of olive leaf supplementation on the constituents of blood and oxidative stability of red blood cells.

This was most likely a factorial design. There were significant product x treatment interactions based on SEM of each parameter in all tables. Please check again.

Response: According to the SEM obtained initially we considered that there could be significant differences in the interaction between product and treatment. But contrary to what was expected, these differences were not obtained. This is due to the fact that the standard deviation is high, which makes any significant difference in the interaction between product and treatment impossible. The standard deviation is due to the number of animals used, coming from semi-extensive farms. In particular, we do not consider that a standard deviation in this type of sample is high. We are talking about a sample with a very high variability, coming from animals that are not inbred, which increases the amplitude in the differences between the animals.

The authors did not highlight different varieties of olive cake added to the diets that contributed to the assessed parameters in the results and discussion.

Response: the project focuses on the types of extraction used to obtain olive cake. The objective of this work is not to do a characterization of the varieties used, and to define which are the most appropriate. In this work all the residues from the Trás-os-Montes region (whose varieties are multiple - cobrançosa, madural, verdeal and negrinha de freixo) were collected and from a certain quantity the different extraction methods (which is the focus of this project) were applied. It should be noted that there are a number of small and medium producers that would make any treatment by variety unfeasible, not being high enough to make their extraction advantageous.

The project intends that after its conclusion, there will be a real applicability, and that this will facilitate the disposal of a toxic residue and allow a cost reduction in the feeding of pigs. The projects should be carried out based on its applicability in the real world.

Finally, please be concise and explicit about what you discovered in this work. Because your data showed that treatments had no meaningful impacts, no additional comments were made to the conclusion.

Response: we believe that the conclusion described in this work is in agreement with the results obtained. There were no significant differences, which allows us to conclude that the processed products had no negative or positive impact with the inclusion of the different types of olive cake. This is a positive conclusion in the sense that this toxic residue can be introduced in animal feed, and the final consumer will not have any different perception that could influence the choice of the product. In addition, the olive oil industries will have a way to dispose of their toxic product. We therefore consider that this work will have a real impact on pig producers and on the olive sector.

Round 2

Reviewer 3 Report

Please find it in the attachment.

Author Response

REV.3

Dear Authors, the manuscript is interesting, but I suggest to Editor to reject it in the current form. The manuscript contains many errors and omissions. Detailed analysis below.

  1. Title should be changed – not only olive cakes were in diets but also olive oil, so it is not precisive.

Response: we do not think it is appropriate to change the title of the article. Adding 1% olive oil would only create entropy in the article, adding nothing new to the reader.

I don`t agree. It is not only one type of products called “olive cake” in diets: You have 3 different products: 1. crude olive cake; 2. exhausted olive cake; 3. olive oil. So, title is not fully informative.

2º Rev - Response: based on your comment, we have made the decision to modify the paper's title, believing that this adjustment will enhance its clarity and knowledge value.

2.Abstract

-           Lack of information about differences among groups and about experiment, number of pigs and samples.

Response: Suggestion accepted; changes have been made in the revised version of the manuscript.

I don`t see it in revised manuscript - abstract. There is nothing about diets, pigs and samples.

2º Rev - Response: we have made the requested changes. The changes are between line 26 and line 30

Lack of conclusion.

Response: Suggestion accepted; changes have been made in the revised version of the manuscript.

These findings highlight the triple positive strategy of adding olive cake to the pigs' diet.

“Firstly, it allows for cost reduction in pig feed formulations by incorporating smaller proportions of olive cake, a by-product readily available in large quantities and with minimal residual costs, thus promoting a circular economy. Secondly, the utilization of olive cake opens up new avenues for research, while simultaneously reducing the amount of hazardous and toxic waste generated from olive processing, thereby benefiting the environment”. – Those are not conclusions from your work, because they were not the aims of your work. Where did you in the manuscript calculate it?

2º Rev - Response: we consider focusing our conclusion on what has been achieved in this work. The changes are between line 44 – 46.

3.Introduction

- L 63-65 – these are spices and you must concern on olive cakes – what is special in them? What do you expect and why? What kind of volatile substances were important? What is mechanism of production of volatile substances from diet?

Response: Spices are extremely important in volatile compounds and are key in the development of terpenes and sulfide compounds. These compounds impart flavor to cured products, which makes them more acceptable to consumers. The volatile compounds derived from the oxidation of unsaturated fatty acids, namely through the lipoxygenase pathway, Maillard reactions and the Streaker degradation of some amino acids.

But species are not the aim and no in title of your study. You should concentrate on olive cakes. How can olive cakes impact the product?  Why you use them? What do you expect and why?

2º Rev – Response: Suggestion accepted; changes have been made in the revised version of the manuscript (line 77-89)

  • L 82-86 – the aim should be simple. This description should be in M&M part.

Response: we think that the objective is simple, we just think it is important to mention the name of the treatments, which makes the sentence longer, but this does not mean that the objective is not simple.

I don`t agree. The description of particular diets should be in M and M.

 “So, the aim of this study was to evaluate the effect of the inclusion of different types of olive cakes and olive products in the finishing diet on volatile compounds of dry-cured loin and dry-cured “cachaço” from Bísaro pig”

2º Rev – Response: Suggestion accepted; changes have been made in the revised version of the manuscript (line 158-160)

  • What`s about welfare of pigs? Any standards?

Response: The Bísaro pigs were slaughtered at the Municipal Slaughterhouse of Bragança as described by Álvarez-Rodríguez and Teixeira and in accordance with applicable legislation of welfare of pigs.

I asked about standards in pigs` maintenance, not about action in slaughterhouse. You said that you would like to present this race, but you did not describe pigs` environment. We only know it is extensive production. What is area they live, how many fodder, drinker per group of pigs? How were diets offered? How you measure diets intake? How were groups separated?

2º Rev – Response: This work is part of a project being co-promoted with an extensive Bísara pig farm, along with other research centers. Our work area started at the slaughter of the animals. The whole farm phase (determining the lots, amount of food available per day, access to water, weight gained per day, among others) was the part of action by the other project partners.

  • L 97 - In total, forty animals of the Bísaro breed were used. – not necessary – 5 x 8=40

Response: Eight animals were used for each treatment (control, T1, T2, T3, T4 and T5), making a total of 40 animals. We don't understand the question.

So, look 109-110 and 115-116.

2º Rev – Response: changes have been made in the revised version of the manuscript.

  • 100-107 – it should be in table and 108-109 – where are these results?

Response: We also consider that presenting this information in a table format would be beneficial. However, we have encountered challenges with the repeatability percentage in the past. In previous articles, we included this information in a table; however, for this particular instance, we intend to provide a concise summary along with a bibliographic reference.

“For SFA the values ranged from 20.03 to 21.54 % for all treatments. As for MUFA, they varied between 28.0 and 42.36 %. As for PUFA, the values ranged between 37.61 and 50.46 %. C16:0, C18:1n-9 and C18:2n-6 are the major fatty acids of SFA, MUFA and PUFA, respectively, in the different treatments used for the diet of the Bisaro pig animals. The chemical composition and the fatty acid profile of the diet applied to the animals were made according to Leite et al. [9].”

When I read above, I understand that you use this some method of fatty acids determination as in Leite et al. If this information was published before you should sign it clearly that  “fatty acids profile of used olive cakes was analysed and published before in…(publication)…in table….(number, page). “ – or in similar form as in response:

Response: table 1 in the article: Leite, A.; Domínguez, R.; Vasconcelos, L.; Ferreira, I.; Pereira, E.; Pinheiro, V.; Outor-Monteiro, D.; Rodrigues, S.; Lorenzo, J.M.: Santos, E.M.; Andrés, S.C.; Campagnol, P, C, B.; Teixeira, A. Can the introduction of different olive cakes affect the carcass meat and fat quality of Bísaro pork? Foods, 2022, 11, 1165. https://doi.org/10.3390/foods11111650

 but it should be highlight in revised manuscript.

2º Rev – Response: changes have been made in the revised version of the manuscript.

  1. Materials and Methods
  • M&M section should be carefully sorted! The section is poorly informative.There are no tables with diets and their nutritional values. There is also no basic information about olives cakes and their value.

Response two observation: We disagree with this observation. In fact, we believe that our work contains information that many published articles do not include, such as the manufacturing flowchart of the processed products. Typically, studies related to the industry are limited in terms of providing information about their manufacturing steps. Moreover, we have even included previously published articles with the same animals to provide additional information. Since the materials and methods are practically identical, we may be subject to scrutiny for high repeatability, as has happened before.

  1. So where is information about published before diets value, olive products characteristic, pigs` performance? If in title of this manuscript you concern on diet and pigs it means it is crucial for this study. Cite these works.

2º Rev – Response: After the observations by the reviewer, we decided to put a table with relevant data from the different olive pomace used in this study (TABLE 1).

  • 93 what is “Basic diet and commercial feed” ???

Response: These animals are in a semi-extensive regime, with access to horticultural products from the region (turnips, potatoes, cabbage, fruit, etc.) plus the typical commercial pig diet.

  1. So where is this news in the manuscript?? Short description must be added in manuscript. How can you be sure that these animals consume your prepared diets?

2º Rev – Response: As previously mentioned, this work is part of a co-promoted project. One of our partners was responsible for all the animal part, that is, responsible for separating the animals by lot, applying the olive cake daily, doing the average weight gain weighing, among others. Therefore, this partner together with the swine farm were responsible for the accomplishment of these works. We do not question the excellent work of our partners. Regarding the missing information, we believe that it makes perfect sense, and that it was an oversight on our part.

L 94 what is “Basic diet + 10% olive cake in two phases” - you don`t meaning about any phases in manuscript.

Response: Centrifuged olive cake can exhibit either a biphasic or three-phase nature. It is only the centrifuged olive cake that can possess two or three phases, thus justifying our description in this manner. Once again, articles from our project partners are currently being published, offering clear discussions on the different types of olive cake and providing comprehensive descriptions on the subject.The two-phase system is more ecofriendly, as the olive cake is moister and contains lower oil content, due to the more efficient separation of the oil from the other elements by the centrifugation system. The three-phase olive cake is drier, as during the extraction of olive oil, the olive paste is separated into three parts: olive oil, dry olive cake and olive mill wastewaters. The objective and expertise of our research group do not aim to provide information about olive cake to the reader, but rather to investigate its impacts on processed products and whether its inclusion may lead to rejection by end consumers.

  1. Where is this information in the text about what kind of “phases” do you mean? For me f.e. phase = starter, grower, finisher. This short description must be added in manuscript.

2º Rev – Response: changes have been made in the revised version of the manuscript (abstract and in point 2.1 immediately before table 1)

  • What is the difference between exhausted olive cake and olive cake? What is the characteristic of olive oil?

Response: The olive cakes can also be characterized by their composition and oil content, as crude olive cake and extracted olive cake. The exhausted olive cake is easier to transport, has greater conservation capacity and is available for a longer period of time that olive cake. Regarding the added olive oil, it does not pertain to a single variety of olives, but rather the oil obtained from the processing units in the Trás-os-Montes region.

Great. But where is this information in revised manuscript?

2º Rev – Response: changes have been made in the revised version of the manuscript (In the first paragraph of point 2.1)

  • What mass had the pigs at the start and at the finish? Was one or more phases??

Response: The various treatments were administered to the animals (in batches) during the finishing phase, specifically in the last 90 days prior to their slaughter. Data collection took place daily to assess weight gains and other relevant information. However, once again, these data belong to our project partners, whose publication of the studies is currently in the finalization phase.

The mass of animals is important if we think about meat products, especially from native breed. You should cite the works or give mean value of their initial and final body mass.

2º Rev – Response: These results are known to all members of the project that this work is part of, but have not yet been published (and are in the process of being published). There was a poster at the EAAP congress (a printout follows). We believe that this data should not be published by our research group but by our partners.

  • L 119-125 – differences should be described in introduction.

Response: We do not believe that this information should be included in the introduction in a detailed manner. Instead, it should be presented in the materials and methods section.

It is general information. And it is connected with general characteristic of these products. As you were highlight in Introduction: “Therefore, the present study makes it possible to complete the little existing information about this breed and its processed products” (L 90-92) - but in this moment there is something in Introduction, something in M&M and reader has not full view.

2º Rev – Response: we consider that the changes made (in previous questions) are sufficient to close this issue.

  • Were the differences in meat composition among groups?

Response: there were differences in the chemical composition of the meat. These differences are described in other works that serve as a bibliography for the current work. The works follow:

Leite, A.; Domínguez, R.; Vasconcelos, L.; Ferreira, I.; Pereira, E.; Pinheiro, V.; Outor-Monteiro, D.; Rodrigues, S.; Lorenzo, J.M.: Santos, E.M.; Andrés, S.C.; Campagnol, P, C, B.; Teixeira, A. Can the introduction of different olive cakes affect the carcass meat and fat quality of Bísaro pork? Foods, 2022, 11, 1165. https://doi.org/10.3390/foods11111650

Leite, A.; Vasconcelos, L.; Ferreira, I.; Domínguez, R.; Pereira, E.; Rodrigues, S.; Lorenzo, J. M.; Teixeira, A. Effect of the inclusion of olive cake in the diet on the physicochemical characteristics of dry-cured loin and dry-cured “cachaço” of Bísaro pig. Applied Sci. 2023, 13, 1439. https://doi.org/10.3390/app13031439

This info should be added in text of manuscript. The meat composition was published in…Leite et al. 2022...etc. Reader don`t know if you cite some other publication or a fragment of your previous publication connected with this same research or with other, so you must be precisive.

2º Rev – Response: Changes made to the last paragraph of points 2.1 and 2.2

  • Is the taste of products connected with fat content? Were the differences in this trial?

Response: the amount of fat in both products is very different, being significantly higher in the steaks. The amount of fat present in a product interferes with the evaluation of taste, texture, flavor, chewiness, and other sensory parameters. The issue of fat is referred to throughout the document.

Where is the information about fat content and fatty acids profile in meat? The content of fat and fat profile in product is the result of fat content and profile in meat and this is the result of fat content and profile of diet – so where are these information?

2º Rev – Response: With the changes made for the previous question, we believe that the reader will be able to consult this information more easily.

  • Statistic – I don`t understand why you compare both products together if they are different. They should be analyzed separately.

Response: we were unable to perceive this issue. Both products were evaluated individually, statistically speaking the statistical differences between them were evaluated. The goal of this work was to try to understand if both products were different in terms of volatile composition, and already published articles show that both products are very different in terms of their chemical composition

Look at your title : Did the addition of olive cake in the finishing diet of Bísaro pigs affect the volatile compounds and sensory characteristics of dry-cured loin and dry-cured “cachaço”?

If you compare products it should be “The comparison of volatile compounds and sensory characteristics of dry-cured loin and dry-cured “cachaço achived from Bisaro pigs fed with the addition of olive cakes” .

The products should be compared but every product separately with their 5 variations connected with groups (diets). In this moment is like comparison of yellow cheese with white cheese – both come from milk.

2º Rev – Response: In line 314-315 (point 2.5) it reads as follows: “the effect of product, treatment and the interaction between product x treatment”. We still don't understand what the doubt is at this point.  

5.Tables

- What is LRI in tables??? What is m/z? Groups and short names should be described under the Table 1.

Response: LRI: linear retention index calculated for DB-624 capillary column installed on a gas chromatograph equipped with a mass selective detector; m/z: quantifier ion. Information included in table 1

Really, where? I see it only under table 2.

2º Rev – Response: we agreed, the information was not in the first table. We decided to use a single table (table 2)

  • The titles of tables are not informative and too long – it should be added which groups of substances are in tables.

Response: we don't consider that the information replicated in the table title adds information to the reader. We think this way is much easier and more effective.

Really? Compare!

Table 1. Volatile compounds (expressed as AU 105/g) of dry-cured loin and dry-cured “cachaço”. Effect of treatment with olive cake, product.

Table 2. Volatile compounds (expressed as AU 105/g) of dry-cured loin and dry-cured “cachaço”. Effect of treatment with olive cake, product.

Is informative for you? If tables present this same it should be only one table.

2º Rev – Response: we agree. We decided to use a single table (table 2)

6.Results and discussion

-           Is the correlation or similarity between sensual and chemical analysis?

Response: The sensory evaluation by the tasters is related to the amount of fat in each of the products.

Where is this added in manuscript?

2º Rev – Response: this information is on the line 649-650: “According to Leite et al. [13] the dry-cured “cachaço” has a higher total amount of fat, which will be directly related to the sensory assessment by the panel tasters.”

  • L 420-430 – what is the conclusion from that?

Response: The purpose was to understand if this analysis, taking into account the evaluation by the panel of tasters, could group or not the different treatments. Based on the 16 sensory parameters evaluated, this discriminant was able to separate part of the treatments, that is, there are characteristics that separate them from each other.

But what it means for reader? Where is conclusion of this evaluation added in the revised manuscript?

2º Rev – Response: This information is more important for the manufacturing industry than for the end consumer. This work does not have a single academic perspective. This work, embedded in this project, also aimed to provide information to the manufacturing industry regarding consumer trends for a specific treatment.

7.Conclusion

- Can you identify „sensory analysis and volatile profile as two parameters”?

Response: Yes, but we don't understand the point of the question. These are the two issues addressed in the paper

“Sensory analysis” contains many different parameters, similarly as “volatile profile” – so you can say “two groups of parameters” in my opinion.

2º Rev – Response: we agree. Changes have been made in the third line of conclusion

  • L 438-443 - there is not conclusion from this study – generally no effect of olive cakes was found.

Response: This statement does not seem correct to us at all. Not all studies have to show significant differences to be validated and useful. In our case it is really positive that there are no significant differences in the volatiles profile and the evaluation by the tasters. This shows that the product is not negatively affected by the inclusion of olive pomace in the animals' diet. And that it could therefore be a circular economy option. A product considered very toxic can be part of the animal feed of pigs of the Bisaro breed, without adversely affecting its composition.

Yes, but  you have writen “ These results also show that the addition of olive cake in the pigs’ diet is a triple positive strategy: 1. It is possible to reduce costs with pigs’ diet, by introducing smaller percentages of olive cake (a by-product with residual costs and available in high amounts, promoting in this way the circular economy); 2. The use of olive cake will open a new line of research for its use and will reduce the quantity of this olive waste to process, being highly beneficial for the environment (since the olive cake is a hazardous and toxic by-product)”  - it is not true, because you did not analyze these points! You cannot say that: “these results show” – look also in Abstract.

2º Rev – Response: we agree. Changes have been made in the conclusion.

8.Literature is not prepared in accordance with the requirements.

Response: the bibliography used is based on other authors who have studied volatile compounds in processed products from other pig breeds, the use of by-products in the industry, the sensory evaluation of tasters, etc. We consider that the bibliography used is fully adjusted to our work.

It is not connected with the choice of literature but with presentation of literature according to Journal guide:

Look: short or full journal name? sometimes point on the end, sometimes lack of point – check literature carefully!

Ramírez, R.; Cava, R. Volatile profiles of dry-cured meat products from three different Iberian X Duroc genotypes. J. Agric. Food Chem. 2007, 55, 1923-1931. 10.1021/jf062810l

Lorenzo, J.M.; Carballo, J.; Franco, D. Effect of the inclusion of chestnut in the finishing diet on volatile compounds of dry-cured ham from celta big breed. Journal of Integrative Agriculture. 2013, 12, 2002-2012. https://doi.org/10.1016/s2095-3119(13)60638-3.

2º Rev – Response: were reviewed again

Reviewer 4 Report

Please convey only what is discovered in the work in the abstract; extra ideas should be provided in the main text.

Throughout the manuscript, please change "basic diet" to "basal diet."

In the introduction, the authors did not indicate what flavors from olive cake can influence the preference for meat products, which is a critical aspect. The authors should change the title, which underlined the differences between two items.

 Although I realized that this investigation was followed by a feeding trial, I was perplexed as to why the authors attempted to establish a link between olive cake and volatile chemicals and sensory features in both items.

In my experience, there is a considerable relationship between product and treatment. Please double check.

Author Response

Please convey only what is discovered in the work in the abstract; extra ideas should be provided in the main text.

2º RESPONSE - REV 4: the abstract was changed, trying to accommodate all the suggestions and change requests from the various reviewers.

Throughout the manuscript, please change "basic diet" to "basal diet."

2º RESPONSE - REV 4: thank you for the suggestion. We have made the change in the text of the article.

In the introduction, the authors did not indicate what flavors from olive cake can influence the preference for meat products, which is a critical aspect. The authors should change the title, which underlined the differences between two items.

2º RESPONSE - REV 4: thank you for the suggestion. We have made the change in the text of the article.

Although I realized that this investigation was followed by a feeding trial, I was perplexed as to why the authors attempted to establish a link between olive cake and volatile chemicals and sensory features in both items.

2º RESPONSE - REV 4: We understand your comment but, as mentioned by yourself, this work is part of a greater scientific project which link several scientific fields, interconnected with the industrial sphere. Besides the intention to evaluate animal physiology, meat quality, processed pig products and some industrial parameters, we also intend to go beyond that, and with this we intend to reach the consumer. The addition of olive by-products to pig’s diet, in this case olive cakes, also introduces a substantial amount of fat (from the olive oil remaining in the olive cake). Olive oil fat is characterized for being mainly monounsaturated with appreciable amounts of polyunsaturated fatty acids. As you may know, part of the volatile compounds formed in fatty food products came from the lipoxygenase pathway (LOX), a metabolic pathway that involve fatty acids (mainly polyunsaturated ones) and specific enzymes that yield hydroperoxides, leading to the formation of volatile aldehydes, alcohols, ketones, and other volatiles. The inclusion of olive cakes, still with olive oil that has not been able to be extracted during mechanical and physical extraction, could influence the amount of animal’s fat and the composition of that fat. Due to this fact, the processing of meat products, with appreciable amounts of animal fat, could induce to changes in the volatile fraction. In extreme conditions, the appearance of off flavors could be an hypothesis, such as rancid, from the LOX pathway. Since consumers acceptability is important, we intended to verify if the volatile composition was not severely affected in a way that could lead to consumers rejection or the detection of off flavors. This information is important form the academic point of view, but it is even more important for industrials and for their commercial strategies for the creation of innovative meat processed products. Therefore, our ultimate intention with this work was to create a possibility of valuing if the introduction of olive cakes could benefit or at least not prejudice the volatile composition and consumers acceptability of these new products.

In my experience, there is a considerable relationship between product and treatment. Please double check.

2º RESPONSE - REV 4: According to the statistics performed, there are no significant differences in the interaction between treatment and product. Once again, we inform that the standard deviation is high, which could be behind these results.
